# SlowITe, a Novel Denial of Service Attack Affecting MQTT

**DOI:** 10.3390/s20102932

**Published:** 2020-05-21

**Authors:** Ivan Vaccari, Maurizio Aiello, Enrico Cambiaso

**Affiliations:** 1Consiglio Nazionale delle Ricerche (CNR), IEIIT Institute, 16149 Genoa, Italy; maurizio.aiello@ieiit.cnr.it (M.A.); enrico.cambiaso@ieiit.cnr.it (E.C.); 2Department of Informatics, Bioengineering, Robotics and System Engineering (DIBRIS), University of Genoa, 16145 Genoa, Italy

**Keywords:** internet of things, protocols security, cyber-security, network security, slow dos attack, mqtt

## Abstract

Security of the Internet of Things is a crucial topic, due to the criticality of the networks and the sensitivity of exchanged data. In this paper, we target the Message Queue Telemetry Transport (MQTT) protocol used in IoT environments for communication between IoT devices. We exploit a specific weakness of MQTT which was identified during our research, allowing the client to configure the behavior of the server. In order to validate the possibility to exploit such vulnerability, we propose SlowITe, a novel low-rate denial of service attack aimed to target MQTT through low-rate techniques. We validate SlowITe against real MQTT services, considering both plain text and encrypted communications and comparing the effects of the threat when targeting different daemons. Results show that the attack is successful and it is able to exploit the identified vulnerability to lead a DoS on the victim with limited attack resources.

## 1. Introduction

Nowadays, the Internet of Things (IoT) is a consolidated technology and it is currently adopted in many contexts and applications of different nature. In the Internet of Things environments, simple  objects are able to manage, process and communicate data of the surrounding environment and send them to other IoT devices or to more complex systems. In the IoT world, people and objects can directly interact with each other, thanks to the spread of smartphones, tablets and other mobile devices that provide Internet access from anywhere. Because of the rapid developments in the underlying technologies, IoT introduces new opportunities for a large number of applications, potentially able to improve the quality of human life. IoT networks and devices are indeed widely adopted in different scenarios such as home automation, Industry 4.0, healthcare and critical infrastructure domains: through IoT devices and networks, applications can remotely monitor environments, or to control more complex systems such as smart light bulbs, robotics, health parameters, sensor networks and smart integrated systems. In particular, IoT is a widely adopted in Industry 4.0, where intelligent devices and objects are revolutionizing the business and production scenario, increasingly connection and optimization between production machines through digital data and analysis. Industrial Internet of Things (IIoT) is a term conceived and developed to be adopted and applied exclusively in the context of the fourth generation industry [1]. The purpose of IIoT is to optimize production processes, by connecting machines together and supporting data processing to allow predictive analytics activities, able for instance to predict maintenance requirements, hence potentially leading to significant cost reductions and to more efficient production systems.

An ad-hoc communication protocol available for implementing IoT networks is Message Queue Telemetry Transport (MQTT), a publisher/subscriber system introduced in 1999 [2] and adopted nowadays  [3] in both IoT [4] and IIoT [5] contexts. The MQTT protocol is based on the TCP/IP stack and it is positioned at the application layer of such stack. Being a protocol adopted in IoT and IIoT networks, communications security is a critical and delicate aspect, since processed data/information are related to sensitive industrial contexts [6].

In this paper, we focus on the analysis and study of the security of the MQTT protocol, since such protocol is adopted both in industrial [7] and home automation contexts [8], with the aim of identifying vulnerabilities and design new attacks methodologies exploiting the weaknesses of the protocol. Particularly, the proposed work is contextualized in the IoT security topic, where we introduce the exploitation of a specific parameter of MQTT, through the design, development and validation of a novel threat known as SlowITe, acronym of Slow DoS against Internet of Things Environments, and targeting MQTT services. The proposed attack is a denial of service (DoS) attack aimed to make a network service unavailable to its intended users. SlowITe belongs to the Slow DoS Attack (SDA) category of DoS attacks [9] and it is specifically designed to target the MQTT protocol, adopting the low-rate approach common of other slow DoS threats. Since, to the best of our knowledge, no previous SDA threats are designed to target MQTT, while the main focus are HTTP and HTTPS [10,11,12,13], the proposed SlowITe threat should be considered a relevant advancement in the IoT security field.

The remaining of the paper is structured as follows: Section 2 reports the related work on the topic. Section 3 introduces the MQTT protocol, while Section 4 describes in detail the proposed attack. Executed tests and obtained results are reported in Section 5. Section 6 introduces instead protection approaches able to defend a network system from the proposed threat. Finally, Section 7 concludes the paper and reports further works on the topic.

## 2. Related Work

The starting point to investigate the security of the Internet of Things is to analyze the most adopted communication protocols. In this context, [14,15] review different IoT communication protocols, with a focus on their main features and characteristics. Authors compare 6LoWPAN, acronym of IPv6 over Low-Power Wireless Personal Area Networks, ZigBee, Z-Wave and Bluetooth Low Energy network protocols. Similarly, [16,17,18,19,20,21] investigate both communication protocols such as ZigBee, 6LoWPAN and Z-Wave protocols, also considering application layer protocols like CoAP, MQTT, AMQP. Instead, if we consider investigation of specific protocols, ZigBee security is studied in different works, proposing specific and novel cyber-attacks targeting such protocol [22,23,24,25] or analysing the impact of well-known cyber-threats such as replay attack, sniffing, brute force, flooding or worm [26,27,28,29]. Concerning Z-Wave, several works focus on security of such protocol: [30] identifies a specific vulnerability in the encryption/decryption phase of the protocol, while [31] proposes a cyber-threat able to insert a Rogue controller inside a Z-Wave network. [32] evaluated instead security of Z-Wave networks by adopting different approaches based for instance on Radio Frequency (RF). Another  IoT communication protocol alternative to ZigBee and Z-Wave is 6LoWPAN. Security of 6LoWPAN networks is investigated in [33], analyzing vulnerabilities of the protocol. In addition, investigation of different attacks targeting such protocol is found in literature, including denial of service (DoS) [34], sinkhole [35], fragmentation [36] or wormhole [37] attack. Although the proposed work focuses on the introduction of a novel DoS threat targeting IoT networks, our focus is on the MQTT application layer protocol. In addition, unlike existent threats, the innovative denial of service attack proposed in this work makes use of low-rate techniques to accomplish its purpose, instead of the adoption of a flooding based approach.

In the field of IoT protocols, it is important to distinguish between communication protocols like ZigBee, Z-Wave and 6LoWPAN, and application layer protocols such as CoAP, AMQP and MQTT [16,17,18,19,20,21]. If we focus on works investigating the security of the application layer IoT protocols, ref. [38] compares MQTT, CoAP, AMQP and HTTP protocols. Instead, [39] analyses security aspects of selected IoT frameworks based on MQTT, CoAP, AMQP and HTTP, while [40] investigates CoAP, MQTT  and AMQP by considering reliability, security, and energy consumption features of such protocols. Although such works focus on application layer protocols like MQTT, they do not focus on the weaknesses of the protocol, to be potentially exploited by a malicious user.

As previously mentioned, the proposed work focuses on the MQTT application protocol, used in machine-to-machine (M2M) communications between IoT devices for real-time data analysis applied to different contexts such as home automation, healthcare, logistics, or industry [41]. To the best of our knowledge, research on MQTT security is still limited. Nevertheless, [42,43] show the adoption of the web service Shodan (available at www.shodan.io) to identify vulnerable MQTT brokers publicly accessible on the Internet. Instead, [44,45,46,47] implement a distributed denial of service attack on MQTT. Unlike the proposed SlowITe threat, in this case, flooding based attacks are considered. Hence, a low-rate approach is not considered.

Previous works we have mentioned highlight critical security aspects of IoT protocols, vulnerable to different attacks and exposed to cyber-security issues, also because of the characteristics of IoT environments. As mentioned, the proposed work exploits a weakness of the MQTT protocol, by  introducing a novel attack targeting it through low-rate DoS approaches. Considering the impact of the threat, able to maintain a DoS for long times and by using a very limited amount of resources, the proposed work should be considered a relevant step in the IoT and cyber-security fields, due to the application of low-rate DoS attacks to target IoT protocols like MQTT. In addition, the proposed work should be considered relevant in the network protocols security topic, since it is needed to identify and validate protocols weaknesses, with the final aim of designing more secure protocols.

## 3. The MQTT Protocol

MQ Telemetry Transport (MQTT) is an application layer protocol introduced by IBM in 1999. Although it should not be considered a recent protocol, MQTT is nowadays adopted for different applications such as handling mobility [4], monitoring data [48], notification systems [49], heart and ECG monitoring [50,51], and it was adopted in the past by large-scale companies like Facebook in the Messenger application [3]. In 2016, the Organization for the Advancement of Structured Information Standards (OASIS) declared MQTT as the reference standard for communications on Internet of Things environments [52]. Because of this, MQTT is still considered an important protocol in the IoT field and its security assumes a crucial and critical role.

MQTT is considered a lightweight protocol, since its messages have a reduced code footprint. Each message consists indeed of a fixed header, an optional variable header, a message payload limited to 256 MB, and a quality of service (QoS) payload. Considering MQTT version 3, scope of the proposed work, three different QoS levels are supported and they determine how the content is managed by the MQTT protocol [4]. MQTT stack is composed of three different layers: (i) a physical layer, (ii) a TCP/IP stack, and (iii) an MQTT application layer.

The protocol adopts a publish/subscribe communication model based on a central node hosting the MQTT server called a broker. MQTT is used for M2M communication and, because of this, it has an important role in the Internet of Things [53]. By considering Figure 1, we representing a sample scenario of an MQTT network, clients are able to send or publish information/messages on a given topic, which is a specific virtual channel/room, to a server that acts as the MQTT messages broker. After receiving a message from a client, the broker sends such information to the customers that previously subscribed the same topic.

The MQTT protocol can be used on wireless networks with limited bandwidth constraints or through unreliable connections. Indeed, in MQTT, if the connection between a subscriber and a broker is interrupted, each pending message will be stored by the broker and sent only once the communication is re-established.

An MQTT session is divided into four phases: (i) connection, (ii) authentication, (iii) communication and (iv) termination. A client initiates the connection by creating a TCP/IP connection with the broker on a pre-defined port. Standard ports are 1883 for clear text communications and 8883 for encrypted SSL/TLS communications. Since the MQTT protocol aims to be a protocol for devices with limited resources and IoT sensors [54], SSL/TLS may not always be an option [55]. When SSL/TLS is used, after the connection is established, encryption initiation is accomplished. Hence, the behavior is analogous to the clean text communications case. Authentication can be implemented by sharing, as plain-text, the username and password pair to the server, in a CONNECT packet that represents the first message sent to the MQTT application server. At this point, the server answers with a CONNECT+ACK (CONNACK in the following) packet [56]. Such packets flow allows the MQTT server to authenticate a (client) node on the broker.

## 4. The SlowITe Attack

The Slow DoS against Internet of Things Environments (SlowITe) attack is a novel denial of service threat targeting the MQTT protocol. Such threat belongs to the category of Slow DoS Attacks, making  use of minimum attack bandwidth and resources to target a network service executing a denial of service [9,57,58]. Since Slow DoS Attacks are able to target TCP-based protocols only [9], the MQTT service is considered as a target by SlowITe since running over TCP. Particularly, the aim of SlowITe is to instantiate a high number of connections with the server, in order to seize all available connections the MQTT broker is able to manage simultaneously. Once all available connections (at the application layer) are established by the attacker, the DoS state is reached. At this point, the aim of the attacker is to keep the MQTT broker busy as long as possible, hence maintaining the DoS state, by using at the same time the least possible bandwidth.

In particular, considering a single connection, once the three-way-handshake is accomplished, it is important to initiate a communication with the server, in order to seize the connection with the listening daemon. For instance, for other SDA like Slowloris, this is done in HTTP by sending the beginning of the GET request to the server [9,59]. Such payload can’t be adopted by SlowITe, since the payload accepted by the MQTT server is not compliant to the HTTP protocol. Nevertheless, as previously mentioned, in order to authenticate the client to the MQTT broker, the CONNECT packet is needed. Hence, it is required by the attack to send such a packet to the server. Indeed, SlowITe exploits the CONNECT packet supported by MQTT to instantiate the communication with the broker. The application layer payload of such packet is sized around $s_{1}=30$ bytes (the exchange of username and password pairs is not considered here), depending on the size of the client identifier used to uniquely identify the connection, at the application layer. Once such a packet is received by the server, a CONNACK acknowledgment packet is sent back to the client, with an application payload sized $s_{2}=2$ bytes. At this point, the attacker can follow two different approaches: (i) from one side, avoid sending additional messages to the server, hence, waiting for the server-side connection closure; or (ii) exploit additional packets and communications to keep connections alive for an indefinite time.

Although it may not seem to be the best option, for the implementation of SlowITe, we followed the first approach, since the second approach is not needed in practice, as connections are closed by the server after extremely long times.

In particular, after the CONNECT packet is sent to the server, the server has to keep the connection alive for a period of time equal to k=1.5 times [60] the Keep-Alive parameter *T* , that is by default equal to $T^{\prime }=60$ s [61,62]. This means that by sending a single CONNECT packet it is possible to keep the connection alive for $k\cdot T^{\prime }=T_{DoS}^{\prime }=90$ s. Although similar exploitation on HTTP allows the attacker to keep connections alive for 300 s [59], found value should be considered relatively high, since it means that each connection requires (at the application layer) just $\frac{s_{1}+s_{2}}{8\cdot T_{DoS}^{\prime }}=2.84$ bps. After the Keep-Alive timeout expired, as expected, the connection is closed by the server and the attacker needs to establish a new connection with the server in order to seize the freed connection slot again.

Although such approach may be successful, from the attacker point of view, as it would be able to potentially size all connections available on the server, hence reaching the DoS, and keeping the server unavailable for around $T_{DoS}^{\prime }=90$ s, using low-rate techniques, SlowITe exploits a specific vulnerability of MQTT that makes the attacker able to set the Keep-Alive parameter to an arbitrary value. Such openness of the protocol should be considered a relevant weakness of the MQTT protocol, found during our research and exploited by SlowITe. In order to exploit it, it is possible to specify the value of *T* on the CONNECT packet itself. Hence, it is possible for the client to configure the behavior of the server, in terms of the expiration of the timeouts used for connection closures. Being 16 bits allocated to the value of *T* on the MQTT CONNECT packet, although it is possible to void the Keep-Alive mechanism by setting up a $T_{min}=0$ Keep-Alive value, such configuration may not be allowed by the server: for instance, HiveMQ provides the ability to disable unlimited Keep-Alive values (see https://www.hivemq.com/docs/hivemq/4.3/user-guide/configuration.html). Nevertheless, it is possible to specify a maximum value of *T* equal to $T_{m}=$ 65,535. In this way, once the CONNECT packet is sent to the server, the connection will not be closed by the server, before $k\cdot T_{m}$ seconds. Hence, the connection will be kept alive for around 27 h and 18 min. This means that for such a period of time, no network packets are exchanged between the client and the server, at the application layer. Such extremely high value should be considered an important result of the proposed  work.

The behavior is replicated for a number of connections depending on the server’s configuration and its load. Indeed, in order to make the attack effective and make the MQTT broker unreachable by legitimate clients, SlowITe must seize all available connections. The number of connections that the MQTT broker is able to manage is configured on the running server: considering for instance Eclipse Mosquitto v1.6.2, one of the most known MQTT servers [63], “typically, the default maximum number of connections possible is around 1024” (more details are available on the GitHub source code page of Mosquitto at https://github.com/eclipse/mosquitto/blob/master/mosquitto.conf). This means that with a number of connections equal to $N_{m}=1024$ or greater, it is theoretically possible to lead a DoS on an MQTT server.

Since SlowITe’s aim is to seize all connections of the server through low-rate techniques, the impact on the server, in terms of bandwidth, CPU, or memory is negligible. This is a common characteristic of Slow DoS Attacks, but it should be considered important, since it makes it more difficult to detect running attacks [64]. The proposed attack should, therefore, be considered an innovative threat exploiting a vulnerability of the MQTT protocol through low-rate approaches. In the next sections, we  are going to describe the tests we have executed, in order to validate SlowITe.

## 5. Executed Tests

In this section of the paper, we first describe the adopted testbed, hence report the results we have obtained during the execution of the SlowITe tests. Initially, we validated the SlowITe attack against Eclipse Mosquitto [63] MQTT server, for both plain text and SSL ports. Then, we tested the attack against other MQTT services such as ActiveMQ [65], HiveMQ [66] and VerneMQ [67], comparing obtained results in terms of efficacy and bandwidth requirements.

### 5.1. Testbed

In order to validate the proposed SlowITe attack, a real network has been adopted. The victim host was represented by a physical host based on Ubuntu Linux 18.04 server LTS, running Eclipse Mosquitto v1.6.2 [63] based on the MQTT v3.1.1 and listening on ports 1883 (plain text) and 8883 (SSL). Other scenarios involve tests on ActiveMQ v5.15.12317, HiveMQ v4.3.2 (in its free version) and VerneMQ v1.10.2, running on plain text port 1883 . The default configuration was adopted for each tool. Instead, the attacker is composed of a Raspberry PI 3 Model B based on Raspbian 9.8 and running the SlowITe software over the Java environment. Connectivity between the two nodes is provided through Ethernet connection. The choice to use a low-powered node as the attacker is driven by the characteristics of Slow DoS Attacks, requiring limited resources to accomplish their purpose [9,59,68,69]. It should also be noted that, as previously mentioned, like for other kinds of SDA [57], the  impact of the attack on the server is limited, in terms of bandwidth, CPU and memory. In addition, the Raspberry PI often being used in IoT contexts, we try to execute the attack from an IoT node.

Once the attack is running, we validated it by checking if the DoS is effectively reached on the server. Nevertheless, considering Eclipse Mosquitto (a similar concept is also valid for the other daemons tested), since the number of simultaneous connections the server is able to manage is not well defined (see the “around 1024” limit introduced above), it is needed to check application layer connectivity with a listening daemon during the attack. In particular, a checking node that periodically checks the status of the server is needed, trying to connect to the MQTT broker through a single legitimate connection. Similarly to the concept behind the Schrödinger’s cat paradox [70], such a check connection may alter the status of the server itself. Indeed, such a connection may itself induce a DoS on the server. Nevertheless, our aim is to verify if the attack is able to generate the DoS itself or not. Such verification is accomplished by simulating a legitimate connection and checking if a (legitimate) client is able to connect to the server. Particularly, if the connection is accomplished, the DoS state is not reached. Otherwise, the attack is considered successful.

In order to validate the attack, during its execution, connectivity with the server is checked by the victim itself, by repeatedly trying to connect to the MQTT broker, as a client, every $T_{check}=1$ s, and  checking if the application layer connection is established or not. According to the approach described above, in case the connection with the broker is established, the DoS is not reached. Conversely, if the MQTT connection is not established, the attack is successful.

### 5.2. Obtained Results

In order to test if SlowITe is successful, we first needed to identify the connection closure time, by  analyzing the behavior of a single malicious connection with the MQTT broker. Hence, we needed to validate if the attack is successful, by creating multiple malicious connections with the server and observing the behavior of the server. Finally, we validated the attack on the SSL listening port of the server. Considering such tests, in the following, we report the results we have obtained.

#### 5.2.1. Connection Closure Tests

After a connection is established, according to Section 4, we need to validate such connection is closed after k=1.5 times the Keep-Alive value *T* . Hence, according to Section 4, for default values, being $T^{\prime }=60$ s, we expect a connection closure after around $T_{DoS}^{\prime }=90$ s. In order to test it, we established a single SlowITe connection with the server and observed the behavior of the MQTT broker. We found that the CONNECT command is sent by SlowITe after 0.003 s the send of the first SYN packet. As a consequence, the CONNACK message is sent almost instantly by the server. Then, at time 90.625 the server sends a TCP FIN packet to the client to close the communication, which is definitely closed at time 90.670. Therefore, we found that $T_{DoS}^{\prime }$ is equal in practice to 90.670, that is in line with our expectations. The overall attack bandwidth is equal to 86.29 bps.

In order to analyze the effect of the extended Keep-Alive parameter, we tried to specify it, from the attacker’s point of view, to its maximum $T_{m}=$ 65,535 value. According to Section 4, such value allows us to evaluate the actual possibility to keep connections alive for extremely long times, around 27 h. Similarly to the previous case with a Keep-Alive equal to $T^{\prime }=60$ s, as expected, we have obtained a connection closure after 98,302.876 s, hence, as, expected, more than 27 h. This means that a malicious user could set the Keep-Alive to its maximum $T_{m}$ value, in order to keep connections alive as long as possible and, at the same time, reduce bandwidth requirement to minimum values.

Hence, we have found that with both the configurations adopted, it is potentially possible to lead a DoS on the server, when trying to seize all available connection of the broker. For the next tests, since we have found that the adopted *T* does not influence the possibility to lead a DoS, we have adopted $T=T^{\prime }=60$ s, that, as reported in Section 4, represents the default value.

#### 5.2.2. Multiple Connections Tests

In order to test if the attack is successful and the DoS is effectively reached on the targeted server, it is important to define *N*, the number of connections to establish with the server. In Section 4 we have introduced that such number is “around 1024”. Because of this, in order to identify the number of connections needed for our tests, we used $N=N_{m}=1024$ and executed a SlowITe attack against the MQTT broker service, by targeting the 1883 port, used for plain text communications. At this point, assuming the application daemon is able to manage exactly $N_{m}-N_{e}\leq N_{s}\leq N_{m}+N_{e}$ simultaneous connections, with $N_{e}$ unknown, we expect that one of the following cases is satisfied: (i) $N<N_{s}$, hence  the attack is unsuccessful, as legitimate clients are able to connect the broker; or (ii) $N\geq N_{s}$, hence the DoS is reached, as some of the malicious connections are still not established with the broker. In the latter case, considering an attack creating $N=N_{m}$ connections with the victim, assuming no legitimate connections are established during the observation, we define $N_{e}=\left|N_{m}-N_{s}\right|$. In particular, $N_{e}$ represents the number of connections not exploited by the attack, if $N_{s}\leq N_{m}$. Otherwise, if $N_{s}>N_{m}$, $N_{e}$ represents the number of additional connections that the attack would need to lead a DoS on the victim. Results of the executed tests are shown in Figure 2, by focusing on different time intervals of the attack. In particular, Figure 2a highlights the first 20 s of the attack, where SlowITe instantiates the connections in order to reach the DoS state. Similarly, Figure 2b reports instead details on the time interval where connections are closed by the server, to analyze if, after 1.5 times the Keep-Alive value sent from the client, connections are closed. Finally, Figure 2c shows the entire time scale focused on the maximum number of connections established by the client, to analyze the DoS state during the entire execution of the attack.

Results show that just after 3 s from the beginning of the attack, a large number of connections with the server are established by SlowITe. Particularly, we found that the number of simultaneous connections managed by the server is equal to 1012 . Hence, we found $N_{s}=1012\Rightarrow N_{e}=12$.

By instead analyzing the DoS state on the victim, we found that the DoS is reached as soon as $N_{s}$ connections are established, and maintained until the first connection is closed by the server. Similarly  to the tests reported in Section 5.2.1, such event occurs after around $T_{DoS}^{\prime }=90$ s. In virtue of such a result, we found that N=1024 is a good choice for the execution of the attacks, although each *N* between 1012 and $N_{m}$ would be good in this case.

Considering the connection closures, we found that the last connection was closed by the server after 92.060 s from the beginning of the capture. Such a result was expected, since it is compliant to the results we obtained and reported in Section 5.2.1. We found that the overall attack bandwidth was measured as 86493.85 bps. Compared to the previous test on a single connection, such result is in line with the expectations, since *N* connections are established with the server, instead of a single one.

#### 5.2.3. Tests on SSL/TLS

Previous tests focus on the exploitation of plain text communications, while we will now target an MQTT service supporting SSL/TLS. We expect in this case a minor increase in bandwidth consumption, compared to the previous tests, since an additional security layer is introduced in the communication flow. Particularly, after running SlowITe, we found that the effect of the server is very similar to the one depicted in Figure 2. Indeed, in this case, the DoS is reached just after 4 s from the beginning of the attack. Moreover, the DoS status is maintained until time 95.797, while the last connection is closed at time 95.887. This means that SlowITe is able to exploit not only plain text communications, but SSL-based communications as well.

Nevertheless, we found that the attack bandwidth required to perpetrate SlowITe is in this case increased. In particular, we found that 341188.81 bps are required for the attacker. As expected, such  an increase of requirement depends on the introduction of the SSL/TLS layer on the communication, requiring an additional exchange of messages between the client and the server.

#### 5.2.4. Additional Tests against Other MQTT Service

After executing tests on Eclipse Mosquitto, we analyzed the impact of SlowITe also on other MQTT services. Indeed, since the aim of SlowITe is to target the MQTT protocol, instead of a single software supporting such protocol, we executed the threat on different services. Particularly, as previously mentioned, we targeted ActiveMQ [65], HiveMQ [66] and VerneMQ [67], running on their default  settings.

Considering ActiveMQ, we found that the maximum number of simultaneous connections the server is able to manage is by default equal to $N_{m}^{a}=1000$ (see https://activemq.apache.org/xml-configuration.html). Therefore, during our tests, the malicious software was adopted to establish $N^{a}=N_{m}^{a}$ connections.

Instead, regarding HiveMQ, during our tests, we targeted the free version of the software, which  supports at most $N_{m}^{h}=25$ simultaneous connections (see https://www.hivemq.com/downloads/). Similarly to the ActiveMQ case, SlowITe was configured in this case to initiate $N^{h}=N_{m}^{h}$ connections with the server. It’s important to consider that in this case, although a lower number of connections is adopted, the DoS state could still be reached, since we expect that connection number $N_{m}^{h}+1$ will not be managed by the server, hence experiencing a denial of service.

Finally, as the number of maximum connections supported by VerneMQ is equal to $N_{m}^{v}=10000$ (see https://docs.vernemq.com/configuration/listeners). As preliminary tests showed us that the service is effectively able to manage a slightly higher number of connections, we considered, in this case, $N^{v}=11000$, hence keeping $N^{v}>N_{m}^{v}$.

Similarly to the SlowITe scenario reported in Section 5.2.2, we adopted T=60 s, while traffic capture refers to an overall duration of 200 s. Tests on each service tool were performed individually. Table 1 reports the effects of the attack against each server, where, according to the definition reported above, we consider $N_{m}$ the maximum number of concurrent connections supported by the server and *N* the number of connections established by the attacker. In addition, we define $T_{c}$ the instant of time, represented as second, the first connection is closed by the server, from the beginning of the capture.

For each considered scenario, the attack was successful: after just a few seconds from the beginning of the attack, all connections are established with the server and the DoS is reached. This means that SlowITe is able to target different services supporting MQTT.

Nevertheless, we found that the VerneMQ service is not affected by a denial of service, when  $N^{v}=N_{m}^{v}=$ 10,000 is considered. Indeed, for the executed tests, we found that the server is effectively able to manage 10,009 simultaneous connections, instead of the 10,000 configured. Although the objective of the test to validate the possibility to lead a DoS on a VerneMQ service is reached, the investigation of the behavior of the server and the relation between the adopted configuration and the number of connections the server is able to effectively manage is the scope of further work on the topic.

During our tests, we also found that bandwidth requirements are in line with expectations and they depend on the number of connections established. Particularly, considering the different number of connections established during the attack, the attack bandwidth required is similar for each scenario and VerneMQ requires more bandwidth, while ActiveMQ requires less bandwidth than the other tested services.

In addition, we found that connections are closed, as expected, around 90 s, which is a value in line with the $T_{DoS}^{\prime }$ parameter previously defined. This is not true for ActiveMQ: we found that connections are closed later than expected, hence maintaining the DoS state for longer times. Under such circumstances, where the KeepAlive parameter is set 60, we found $T_{c}=145$. As such value is very close to $T+T_{DoS}^{\prime }$, we presume that the timeout used by ActiveMQ to implement server-side connection closures is initiated after the number of seconds specified as Keep-Alive is elapsed. The  validation of such a hypothesis is the scope of further work on the topic.

Although we found that different configuration is required to target different services, SlowITe was able to lead a DoS on all tested services, hence demonstrating the ability of the proposed threat to target the MQTT protocol in real scenarios.

## 6. Considerations about Protection from SlowITe

As previously demonstrated, the SlowITe attack can represent an important threat for IoT systems based on the MQTT protocol. For this reason, it is important to properly protect the system from attacks like the proposed one. Although such protection is the scope of further works on the topic, in this section of the manuscript we introduce some approaches that could be adopted to define a protection system potentially able to identify and mitigate a SlowITe attack.

In the context of Slow DoS Attacks, detection and mitigation of such threats is still an open challenge in the research field [64]. Particularly, while it may be trivial to detect and mitigate a single attacking node (by filtering out the source IP address), as Slow DoS Attacks could be distributed to multiple nodes, detection of a distributed attack may not be easy, especially considering real-time requirements [71]. Nevertheless, identification and mitigation of Slow DoS Attacks is proposed in the literature, especially considering anomaly-based intrusion detection systems by making use of statistics  [72], spectral analysis [71], Fourier transform [64], or by defining in detail metrics characterizing such kind of attacks [73].

Other research works focus instead on the detection of “old-style” flooding based DoS attacks on MQTT networks: [74] propose a classification model to develop an IDS by using a dataset containing frames under attacks. [75] develops a detection system for DoS attacks based on a network behavior model. [76] proposes ARTEMIS, a framework based on machine learning techniques and algorithms to detect DoS attack against MQTT networks. [45] implements a mitigation approach to mitigate DoS attacks based on QoS features of MQTT and authentication algorithms available on the protocol. Ref. [77] uses a fuzzy logic-based system to detect the malicious behavior of the node combined with a fuzzy rule interpolation mechanism to detect DoS attacks. [78] develops an Intrusion Detection System based on a threshold packet discarding policy to topics defined on the MQTT broker.

All these scientific works can be used as a starting point for developing a SlowITe attack detection system. Nevertheless, it is important to consider that, unlike other services like HTTP where the behavior of a legitimate node could be easier to model, due to the different applications of MQTT, the  classification and categorization of IoT nodes using MQTT could not be trivial. For instance, while in HTTP the aim of a legitimate client is to retrieve a resource from the service in extremely short times, hence making Slow DoS Attacks deviate such behavior (since, as for SlowITe, connections are kept alive for long times during an attack), in case of MQTT, clients, and their connection may remain active without sending any relevant application payload even for hours. Because of this, detecting SlowITe may not be trivial.

Nevertheless, by analyzing each connection between the client and the server during a SlowITe attack, it is possible to define the time of inactivity $T_{i}$ as the time passing between the send of the CONNACK packet/message from the server and the send of the TCP FIN packet, sent to close the connection after the expiration of the server-side timeout. Considering the the Mosquitto scenario previously considered in Section 5.2, we found that the mean of $T_{i}$ is $\mu_{T_{i}}=90.582332985$ s, very similar to $T_{DoS}^{\prime }$ defined above, while its variance is $\sigma_{T_{i}}=0.076503686$. Such values could be adopted and compared to a legitimate MQTT scenario to potentially identify a running attack. Nevertheless, due to the variety of applications of MQTT, defining an appropriate protection methodology is not trivial, although further work on the topic may be directed on the characterization of a set of legitimate MQTT scenarios, with consequent adoption of the approach proposed in [72], to identify running attacks.

Considering protection from SlowITe, further work on the topic may also be directed to refine the MQTT protocol to improve how the Keep-Alive parameter is managed, in order to avoid exploitation from malicious nodes. For instance, it may possible to disable the possibility to configure the Keep-Alive parameter from the client itself: in this case, by forcing the client to use a particularly low parameter, attack bandwidth requirements would increase. It is possible to extend the protocol to use specific and extremely low timeouts forcing the client to send data just after the CONNECT command is sent. A similar approach is used in HTTP [57].

## 7. Conclusions

In this paper, we have investigated the Internet of Things topic and related protocols. We have focused on the MQTT application protocol, widely used in different IoT contexts [4,5], to identify its exposure to denial of service attacks. We used low-rate DoS approaches [9] to target the protocol, by designing and introducing the SlowITe attack, able to target MQTT services to lead a DoS, by requiring limited attack bandwidth. Particularly, we exploited the possibility to set the Keep-Alive parameter of the server from the client itself, hence configuring the behavior of the server, in terms of connections closure timeouts, from the attacking node. This should be considered a weakness of the MQTT protocol, validated in the proposed work, through the design and implementation of the SlowITe attack. In virtue of this, further work on the topic may be focused on a refinement of the MQTT protocol definition, in order to avoid such behavior. Similarly, additional works on the topic may focus on the definition of detection and mitigation systems to counter the proposed threat, or to model the behavior of legitimate MQTT clients.

We tested the attack on a real network, targeting real MQTT services based on Eclipse Mosquitto [63], ActiveMQ [65], HiveMQ [66] and VerneMQ [67] with an low-powerful node represented by a Raspberry PI. Tests were executed by first analyzing the behavior of the server on a single connection, in order to identify the potential exploitation of SlowITe, hence perpetrating real DoS attacks, based on plain text communication and encrypted communications as well. We have found that, after establishment, a single connection can be kept alive for more than 27 h, without sending any data to the listening daemon. Further work may be directed on a refinement of SlowITe to keep connections alive for an indefinite time, to test both new and first version of SlowITe against other possible brokers running on cloud-based solutions, or to further investigate the behavior of ActiveMQ, VerneMQ or additional services. In addition, since we have focused on the MQTT protocol, further investigations on the topic may consider other application layer IoT protocols like CoAP and AMQP [38].

## Figures and Tables

**Figure 1 sensors-20-02932-f001:**
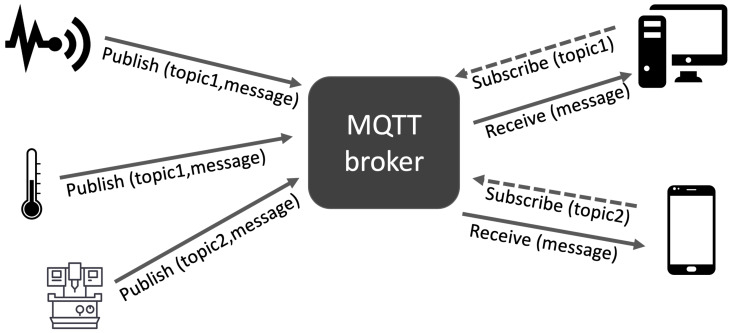
Network sample representation of the Message Queue Telemetry Transport (MQTT) publish/subscribe approach.

**Figure 2 sensors-20-02932-f002:**
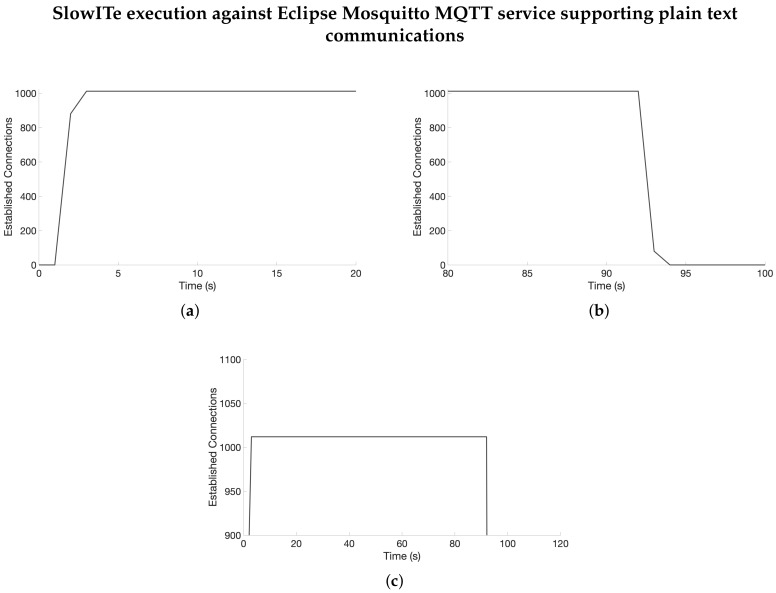
Effects of a SlowITe attack against an Eclipse Mosquitto MQTT service. (**a**) First 20 s of the attack. (**b**) Temporal window reporting connections closures. (**c**) Analysis of the DoS state on the server.

**Table 1 sensors-20-02932-t001:** Comparison between MQTT services.

Targeted Service	$N_{m}$	*N*	$T_{c}$	Network Bandwidth (bps)	
				Total	For Each Connection
Mosquitto	1024	1024	92	38,890.88	38.43
ActiveMQ	1000	1000	145	38,320.96	38.32
HiveMQ	25	25	90	958.96	38.36
VerneMQ	10,000	11,000	90	449,321.92	44.89

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
