# Peer review of "SlowITe, a Novel Denial of Service Attack Affecting MQTT"

_sensors, 2020, doi:10.3390/s20102932_

Round 1

Reviewer 1 Report

The article presents an important topic in IoT security. The SlowITe attack is real. The authors did a good job to demonstrate it and test it.

The testing is done on Eclipse 196 Mosquitto v1.6.2. It is not clear if the vulnerabilities also exit in other MQTT servers. The paper can be further improved by analyzing the impact of the attacks in other MQTT servers, especially, MQTT cloud services.

The organization of the paper is done well. However, the authors need to proof read the paper, check the grammar, and improve the writing of the paper, examples of the changes include, but are not limited to, 

Page 1 line 1: correct email address
Page 1 line 30: change "costs reductions" to "cost reductions"
Page 2 line 73: change "our focus in" to "our focus is"

Reviewer 2 Report

In this paper, the authors presented SlowITe, a novel low-rate denial of service attack aimed to target MQTT through low-rate techniques, which was a denial of service (DoS) attack aimed to make a network service unavailable to its intended users. Compared with previous research, the breakthrough of this attack lies in the analysis and study of the security of the MQTT and meaningfully, according to this paper, it’s the first SDA threat which is designed to target MQTT. This is an innovative thought worth encouraging. In general, the author provides a more detailed textual explanation of some concepts. At the same time, the author proposes a brief reflection on some possible future work in section 6. However, the paper seems to lack the feeling of combining pictures and texts, and it might be more fascinating to add pictures appropriately for explanation. In addition, there are actually some English errors and even most of them are low level mistakes. This reflects the poor writing and careless attitude.

Therefore, this paper should be rejected and needs to be carefully modified for higher quality.

  • In ABSTRACT, it may be better to delete “domains” in “Security of Internet of Things domains is a crucial topic”.
  • In ABSTRACT, attributive clauses are used chaotically, “which was” should be added between “We exploit a specific weakness of3 MQTT” and “identified during our research”.
  • In section 1, “also” in “people and objects can directly interact with each other, also thanks to the” should be deleted.
  • In ABSTRACT, “,”‘ should be added between “In this paper” and “ we have investigated ”.

There are many similar inadequacies to the above in the current version, please check it carefully before next submission.

Reviewer 3 Report

The authors present how to take advantage of the specific MQTT weakness that allows the client to configure server behavior. By exploiting this weakness, an attacker can force a server to keep alive the connection even more than 27 hours.  This relatively simple way of extending the time makes it easy to exceed the maximum number of open MQTT server connections even by inefficient Internet of Things devices, which leads to denied of service  for a long time.

The research confirms the possibility of attacking the MQTT server using this weakness. I am a little disappointed with the content of section 6, the title of which may have suggested that the authors of the paper, after testing, will provide a detailed procedure for preventing such attacks. However, in section 6 you can find suggestions for solving the problem presented in other studies.

I have some insights and comments. To help the reader and to improve
the quality of the manuscript I suggest to modify/consider the
following aspects:

l.5 - It is nowhere explained where the SlowITe acronym comes from

l.58 - The full content of the 6LoWPAN acronym should be given the first time it is used, and not on line 69

l.277 - The font used in the descriptions in Figure 2 is too large

l.277 - The graph in Figure 2 shows the results obtained very poorly due to the scale used on the axes. I suggest presenting the results on three charts. The first chart should show the part of the chart covering time between 0 and 20 seconds, and the second chart should cover time between 80 and 100 seconds. The third graph should show the entire time scale, but the scale of the axis "Established Connections" should cover the range from 900 to 1100. Then the relevant results will be shown quite accurately.

Round 2

Reviewer 2 Report

I checked it thoughout this revision and found no problem any more.